# Incorporation of *Moringa oleifera* Leaf Extract in Yoghurts to Mitigate Children’s Malnutrition in Developing Countries

**DOI:** 10.3390/molecules28062526

**Published:** 2023-03-09

**Authors:** Sandra M. Gomes, Anabela Leitão, Arminda Alves, Lúcia Santos

**Affiliations:** 1LEPABE—Laboratory for Process Engineering, Environment, Biotechnology and Energy, Faculty of Engineering, University of Porto, Rua Dr. Roberto Frias, 4200-465 Porto, Portugal; 2ALiCE—Associate Laboratory in Chemical Engineering, Faculty of Engineering, University of Porto, Rua Dr. Roberto Frias, 4200-465 Porto, Portugal; 3LESRA—Laboratory for Separation Engineering, Chemical Reaction and Environment, Faculty of Engineering, University of Agostinho Neto, Edificio CNIC, Avenida Ho Chi Min 201, Luanda P.O. Box 815, Angola

**Keywords:** *M. oleifera*, bioactive compounds, antioxidant activity, functional foods, yoghurt

## Abstract

*Moringa oleifera*, which is rich in bioactive compounds, has numerous biological activities and is a powerful source of antioxidants and nutrients. Therefore, *M. oleifera* can be incorporated into food to mitigate children’s malnutrition. In this work, the bioactive compounds were extracted from *M. oleifera* leaf powder by ultrasound-assisted solid-liquid extraction. The antioxidant and antimicrobial activities and the phenolic composition of the extract were evaluated. The extract presented a total phenolic content of 54.5 ± 16.8 mg gallic acid equivalents/g and IC_50_ values of 133.4 ± 12.3 mg/L for DPPH and 60.0 ± 9.9 mg/L for ABTS. Catechin, chlorogenic acid, and epicatechin were the main phenolics identified by HPLC-DAD. The obtained extract and *M. oleifera* leaf powder were incorporated into yoghurts and their physicochemical and biological properties were studied. The incorporation of *M. oleifera* did not impair the yoghurts’ stability over eight weeks when compared to both negative and positive controls. The extract presented higher stability regarding syneresis but lower stability regarding TPC compared to the powder. Also, the fortified yoghurts presented higher antioxidant properties than the negative control. These findings highlight the potential use of *M. oleifera* powder and extract as natural additives to produce fortified foods that can be used in the mitigation of malnutrition.

## 1. Introduction

*Moringa oleifera*, a plant from the Moringaceae family, has gained attention all over the world due to its nutritional and bioactive properties. This tree is native to India and grows in tropical and subtropical environments [1]. Most parts of *M. oleifera* are edible and they provide numerous biological activities, such as antioxidant, antimicrobial, anti-inflammatory, antidiabetic, anticarcinogenic, hepatoprotective, and cardioprotective effects [2,3,4,5,6,7,8]. These therapeutic activities can be attributed to a diverse group of bioactive compounds present in *M. oleifera*, such as phenolic compounds and carotenoids. The leaves, in particular, are rich in phenolic compounds, which are secondary metabolites of plants known for their antioxidant properties. These properties can be attributed to the phenolic group (Figure 1) [9]. The most common phenolics found in *M. oleifera* leaves are flavonoids (e.g., epicatechin, catechin, quercetin, kaempferol) and phenolic acids (e.g., gallic acid, chlorogenic acid, caffeic acid) [10,11,12]. Other families of polyphenols, such as lignans, can also be found in *M. oleifera* leaves [13]. Phenolic compounds can be extracted from *M. oleifera* using polar solvents, such as water and ethanol, since these compounds are relatively polar. More specifically, 50% and 70% aqueous ethanol have shown to be more efficient in extracting phenolic compounds from different plant parts [14,15,16,17,18,19]. The phenolic content found in *M. oleifera* leaves’ extracts varies with the origin of the plant, as well as the extraction conditions. Therefore, the values reported in the literature for the total phenolic content (TPC) of these extracts can vary from 4–700 mg of gallic acid equivalents (GAE)/g of dried extract [10,20].

In addition, *M. oleifera* can serve as a supplement against malnutrition since it is a powerful source of nutritional compounds, such as proteins, fats, carbohydrates, fibre, vitamins, and minerals. The most common vitamins in *M. oleifera* are B1, B2, B3, C, and E, while minerals include calcium, phosphorous, and potassium. Compared to other commonly consumed food, *M. oleifera* presents 9 times more protein than yoghurt, 17 times more calcium than milk, 25 times more iron than spinach, and 15 times more potassium than bananas. Also, it contains 10 times more vitamin A and 7 times more vitamin C than carrots and oranges, respectively [21,22]. 

Several in vitro and in vivo studies were performed to analyse the toxicity of *M. oleifera* leaf powder and extracts. Experiments with normal human cell lines demonstrated that the extracts from the leaves are generally safe; however, some cytotoxicity can be observed depending on the dose [1]. One study showed that aqueous leaf extract with concentrations superior to 20 mg/mL was cytotoxic to human peripheral blood mononuclear cells [23]. In another study, ethanolic leaf extracts were considered safe for fibroblasts at concentrations ranging from 0.02 μg/mL to 100 μg/mL [24]. In vivo studies using rats demonstrated the safety of *M. oleifera* leaf powder and extracts [1], which only presented acute toxicity in extremely high dosages (3000 mg/kg) [23]. Finally, no adverse effects were reported in any human studies using *M. oleifera* leaf powder. There are still no studies on humans regarding leaf extracts [25]. Therefore, toxicity studies are fundamental to ensure that the use of supplements derived from *M. oleifera* leaves are safe for human health.

Due to its broad spectrum of biological activities, *M. oleifera* can be used in food, pharmaceutical, and cosmetical applications [10,26,27,28,29,30,31,32,33,34,35,36,37,38]. By combining its extraordinary nutritional value with the presence of bioactive compounds, such as phenolic compounds, *M. oleifera*, more particularly its leaves, is a good candidate to produce fortified food to mitigate children’s malnutrition in developing countries (Figure 2). 

Dairy products are consumed worldwide. They are mostly ingested in the form of fresh dairy products, including yoghurt, especially in developing countries [39]. However, since these products present considerable amounts of fats, they are prone to lipid oxidation [40]. This process leads to the formation of secondary metabolites that can produce an unpleasant flavour and diminish the nutritional properties of the product, decreasing its shelf-life [41]. Since *M. oleifera* leaves are rich in phenolic compounds with antioxidant properties, they can be a natural source of antioxidants to be used in the fortification of food products, increasing their shelf-life (Figure 1). Also, antioxidant compounds are particularly valuable to human health since they can protect the cells against free radicals and reduce oxidative stress, which may reduce the risk of developing certain diseases, such as cancer and heart disease [42]. Numerous studies have shown the potential use of *M. oleifera* as a natural food additive to improve dairy products’ properties and nutritional value (Table 1). 

Although several studies explored the incorporation of *M. oleifera* in dairy products’ properties, very few analysed the antioxidant properties of yoghurts fortified with *M. oleifera* leaves, particularly the leaf powder. Also, the literature review produced no papers that analyse the phenolic composition of the *M. oleifera*-derived ingredients added to dairy products. Therefore, this study aimed to produce fortified yoghurts that can be used to mitigate malnutrition. The study evaluated the effect of *M. oleifera* leaf powder and extract on the stability of the yoghurt, as well as its antioxidant and antimicrobial activities. Moreover, the main phenolic compounds present in *M*. *oleifera* extract were quantified to match the extract’s biological properties to its phenolic composition. Finally, the potential use of *M. oleifera* as a natural food ingredient was assessed and compared to a synthetic additive. 

## 2. Results and Discussion

### 2.1. Extraction of Bioactive Compounds from M. oleifera Leaves

In the present study, the phenolic compounds from *M. oleifera* leaf powder were extracted using 70% ethanol. The extraction yield can be influenced by numerous aspects, such as the extraction method, solvent, time, temperature, and the sample’s mass-to-solvent volume ratio [15,17,19,53]. In general, higher extraction times generate higher extraction yields. However, when high temperatures are applied to extract phenolic compounds, longer periods of extraction can impair the extraction yield since these compounds can be degraded by high temperatures [54]. Ultrasounds can also influence the extraction yield of the compounds of interest. Ultrasounds generate cavitation bubbles within the solvent; eventually, the bubbles collapse and disrupt the cell walls of the solid matrix, releasing the bioactive compounds into the liquid phase (i.e., the solvent) [20]. In the same way, the agitation also helps to disrupt the cell walls of the plant material. Therefore, both ultrasounds and agitation can help improve the yield of the extraction. In this study, the extraction yield was 34.1 ± 0.9%. Other literature reports used similar extraction methods to extract bioactive compounds from *M. oleifera*. In one study, mechanical agitation was performed for 24 h at 25 °C, using 80% ethanol to extract the compounds of interest from *M. oleifera* leaf powder. The extraction yield obtained was 45.12%, higher than the one presented in this work. This might be explained by the higher extraction time used (24 h vs. 3 h) [55]. In another study, the extracts from *M. oleifera* were obtained under constant magnetic stirring for 2 h at 25 °C, using 50% ethanol as solvent. Here, the extraction yield obtained was lower than the one of the present work (26.94%), which may indicate that the use of ultrasounds before mechanical agitation may improve the extraction of bioactive compounds [53].

### 2.2. Characterization of M. oleifera Extract

The extract was characterized regarding its total phenolic content (TPC) and antioxidant and antibacterial properties. The obtained results are described in Table 2. 

The *M. oleifera* leaf extract presented a TPC of 54.5 ± 16.8 mg_GAE_/g_dried extract_. Once again, the extraction method and conditions can influence this parameter, as well as the plant origin. For example, the impact of the drying process used in the pre-treatment of *M. oleifera* leaves (freeze-dried, air-dried, sun-dried, or oven-dried) on the phenolic content of the extracts was analysed. The bioactive compounds were extracted for 24 h on an orbital shaker using water as the solvent and it was observed that the TPC was influenced by the drying process, with the freeze-dried technique obtaining a higher content of phenolics (68.75 mg_GAE_/g). The leaves dried in the oven, the same drying process used in our work, presented a TPC of 46.88 mg_GAE_/g, inferior to the value obtained in this paper [12]. These results may indicate that the ultrasound-assisted solid-liquid extraction method used in the present study may be more suitable for the extraction of phenolic compounds. In another work, the effect of the extraction method on the TPC was also analysed. The investigators showed that mechanical agitation for 24 h was more efficient in the phenolics’ extraction than sonication for 30 min three times. The extracts obtained with the agitation method presented higher TPC (74.87 mg_GAE_/g) than the one obtained in our work, but the extraction time was also significantly superior [55]. 

Regarding the antioxidant capacity, the IC_50_ values obtained represent the extract concentration necessary to inhibit 50% of the free radicals (DPPH or ABTS). It is possible to observe, from Table 2, that *M. oleifera* extract presents a higher antioxidant capacity towards ABTS, since a smaller amount of extract is needed to inhibit the radicals to the same extent. This higher capacity to inhibit ABTS compared to DPPH was also described by other researchers [11,55]. Similar IC_50_ values were obtained for DPPH and ABTS in another study (139.60 mg/L and 57.07 mg/L, respectively) where magnetic agitation was performed for 24 h [55]. In our work, *M. oleifera* leaf powder was placed in an ultrasonic bath for 30 min prior to 2.5 h in agitation. The use of ultrasounds reduced the time of mechanical agitation needed to obtain similar DPPH and ABTS inhibitions.

Concerning the antibacterial activity, it was not possible to identify any inhibition halo, which does not mean that the extract cannot inhibit the growth of *E. coli* and *S. aureus* since the halo can be present underneath the disk. Although other studies report the antibacterial effect of *M. oleifera* leaf extract against these microorganisms, the concentrations tested are particularly higher than the ones analysed in this study, which may account for the differences in the results obtained [34,47].

To better understand the biological properties exhibited by the *M. oleifera* extract, an HPLC-DAD analysis was performed to identify and quantify the phenolic compounds present in the extract obtained. From Table 3, it is possible to observe that catechin was the major phenolic compound found in the leaf extract. Other flavonoids, such as epicatechin, quercetin, and kaempferol were also detected. The main phenolic acid present was chlorogenic acid, followed by caffeic acid and gallic acid. Previous studies have already reported the presence of these compounds in *M. oleifera* leaf extract [10,12]. The values obtained in the literature present some variability since the origin and cultivar conditions of *M. oleifera* tree, as well as the extraction method used to extract the phenolics can affect the compounds’ concentration. In one report, quercetin and kaempferol were identified and quantified. The concentrations obtained were 0.07 mg/g for quercetin and 0.03 mg/g for kaempferol, which were very similar to the ones obtained in the present study [20]. However, other studies present higher concentrations of phenolic compounds. In one study, where the seven phenolic compounds were also studied, the concentrations ranged from 19.65 mg/g for kaempferol to 65.83 mg/g for chlorogenic acid [12]. In another study, the concentrations obtained for chlorogenic acid, epicatechin, gallic acid, kaempferol, and quercetin were also superior to the values obtained in our work, but the concentration of catechin was very similar (20.19 mg/g) [56]. The different origins of the plants (Nigeria vs. Angola) may account for the different compositions obtained.

Since phenolic compounds are known for their antioxidant properties; the antioxidant capacity of the extract against free radicals, such as DPPH and ABTS, demonstrated in this study is in accordance with the presence of natural antioxidant compounds in the extract’s matrix like the ones identified by HPLC.

### 2.3. Characterization of Fortified Yoghurts

#### 2.3.1. Physicochemical Properties

Five yoghurt formulations were produced in this work: a negative control with no additives (NC), a positive control with sorbic acid (PC; a synthetic preservative), two formulations with *M. oleifera* leaf extract at different concentrations (1.0 g/L, ME, and 2.0 g/L, ME2) and, finally, a formulation with 2.9 g/L *M. oleifera* leaf powder (MP). From Figure 3, it is possible to observe that a homogenised product was obtained for all formulations, except for MP yoghurt where it was not possible to completely dissolve the powder. Regarding the colour, fortified yoghurts (ME, ME2, and MP) presented slightly beige colours compared to the controls. 

The stability of the yoghurts produced was analysed by evaluating their physicochemical properties such as pH, syneresis, water holding capacity, and viscosity throughout eight weeks. pH can play a major role in food quality and safety and in yoghurts is usually below 4.6 [57]. From Figure 4, it is possible to confirm that the pH values obtained for all formulations are below 4.6. After production, all formulations presented similar pH values, except PC yoghurt. which presented a slightly higher value. Moreover, a reduction in the pH was observed over time for all formulations; this may be explained by the increase in lactic acid that occurs during yoghurt fermentation [58]. The decrease in the pH value during the eight weeks of the study was higher for NC yoghurt and lower for the fortified yoghurts, and in particular MP yoghurt, which did not present statistically different pH values in the first four weeks. Also, higher concentrations of *M. oleifera* extract did not affect the pH of yoghurts since ME and ME2 formulations presented statistically similar pH values throughout the study. Hence, the results reveal that the incorporation of *M. oleifera* did not compromise the pH of the yoghurts. Furthermore, the fortification of the yoghurts with both extract and powder of the leaves of *M. oleifera* seems a promising strategy to improve the pH stability of the product. 

During yoghurt production, whey proteins from milk are denatured and can interact with each other, forming soluble aggregates, or with casein micelles, forming micelles coated with whey protein. The structure of the yoghurt is given by the interactions between k-casein and whey proteins through disulphide and hydrophobic bonds on the surface of casein micelles [59]. Syneresis refers to the release of whey (serum) from the yoghurt matrix, which results in undesirable sensory properties, making the product less appealing to the consumer [60]. Increasing the water holding capacity (WHC) is a possible strategy to decrease the yoghurt’s susceptibility to syneresis [61]. Therefore, the syneresis and WHC of the yoghurts were analysed over time to understand if the fortification impaired their stability. Eight weeks after production (t_3_) all formulations presented slightly higher syneresis compared to the values obtained after production (t_0_), except for MP yoghurt which presented a slight decrease; ME formulation was the one that presented syneresis values more stable along the eight weeks. On the other hand, WHC decreased in the same period of time, except for the ME formulation where WHC values were more stable. For both syneresis and WHC, the values obtained for ME2 formulation were less stable than for ME formulation. The higher phenolic composition of ME2 may explain these results since phenolic compounds can interact with both casein and whey proteins [62], which could have implications on the yoghurt’s structure. Although fortified yoghurts did not present an increase in the WHC, it was observed that the addition of *M. oleifera* to the yoghurts’ matrix was able to diminish the increase of the syneresis over eight weeks. Hence, this preliminary data indicates that the incorporation of the extract and powder obtained from *M. oleifera* leaves in yoghurts did not impair their physicochemical properties. However, these results must be confirmed in future studies.

Lastly, the viscosity of the yoghurts was also analysed. The apparent viscosity of the formulations was determined, with the shear rate varying from 0.01–100 s^−1^. From Figure 5A, it is possible to notice that the addition of *M. oleifera* to the yoghurt’s composition did not alter the viscosity behaviour of the yoghurts right after production, with all formulations presenting very similar viscosity values. Only PC yoghurt presented slightly lower values for shear rates inferior to 20 s^−1^. Eight weeks after the production, it was possible to observe differences in the viscosity values between the formulations and in comparison to the initial results, with MP yoghurt presenting a higher difference over time (Figure 5B). However, for shear rates superior to 20 s^−1^ this was not observed and the values obtained were very similar to all formulations. Despite the variations in the values obtained, the behaviour of the yoghurts remained the same, with apparent viscosity decreasing until shear rate values of approximately 15 s^−1^ were reached and then increasing, converging to the same viscosity value (around 5 × 10^3^ mPa·s) in both timepoints. The obtained values for the consistency index, K, and flow consistency index, n, are presented in Table 4. From these results, it is possible to observe that all formulations can be considered non-Newtonian fluids with a pseudoplastic behaviour since the obtained flow behaviour indexes were inferior to 1. This type of fluid presents a shear-thinning behaviour, meaning that the apparent viscosity of the yoghurts decreases with an increase in the shear rate, which is true for shear rates inferior to 15 s^−1^. Although all formulations presented a similar behaviour after production (t_0_), it was observed that, after eight weeks (t_3_), the incorporation of *M. oleifera* in yoghurts reduced their shear-thinning behaviour (higher flow consistency index values), which can be related to a lower breakage of intramolecular and intermolecular bonds in the yoghurt matrix [63].

The viscosity behaviour of the yoghurts was also analysed as a function of the temperature (Figure 6). When the temperature increased from 2 °C to 25 °C, an increase in the fluidity of the formulations was observed since their viscosity decreased. On the other hand, when the temperature followed the inverse direction (from 25 °C to 2 °C), an increase in viscosity was observed. However, the viscosity values did not return the initial ones, showing that the change in viscosity due to temperature variations is a process that cannot be reversed. Therefore, these changes in the temperature may cause irreversible modifications in the yoghurts’ structure. 

The results obtained from the physicochemical characterization showed that *M. oleifera* extract and powder can be used to fortify yoghurt without significantly affecting their properties and stability. 

#### 2.3.2. Antioxidant and Antibacterial Activities

The biological properties of the yoghurts produced were also analysed. For that, the total phenolic content and the antioxidant and antibacterial activities of all formulations were assessed. The results obtained for the TPC of the yoghurts are presented in Table 5. It is possible to observe that the positive control (PC) did not present statistically higher TPC compared to the negative control (NC). On the other hand, the fortification of the yoghurts with *M. oleifera* significantly increased the phenolic content of the formulations throughout the entire period of the study. This was expected considering the phenolic compounds present in *M. oleifera*. The results also showed that the TPC of all yoghurts decreased over time, but the values were always higher in the fortified yoghurts. This reduction of phenolics during storage was also described in other studies [30,64]. This may be explained by the biotransformation of phenolic compounds by the probiotics present in the yoghurt [65]. Comparing ME to ME2, besides exhibiting higher phenolic content, ME2 presented a slightly lower reduction of the TPC during the eight weeks. Regarding ME and MP, although they presented similar TPC after production, differences were observed after eight weeks, with MP presenting higher phenolic content than ME. These results demonstrated that although a decrease was observed in the TPC of all yoghurts, the reduction in MP was lower and also slower (significant differences were only observed from four to eight weeks).

Regarding the antioxidant properties, it is possible to observe, from Figure 7 that all formulations presented a higher inhibition capacity towards ABTS than DPPH. These results were expected since *M. oleifera* extract exhibited higher antioxidant capacity in the ABTS assay. Also, another study has shown that plain yoghurt is more efficient at inhibiting ABTS radicals than DPPH radicals [66]. Concerning the DPPH assay (Figure 7A), the fortified yoghurts presented a higher inhibition of DPPH during the first two weeks. However, after two weeks, only ME2 yoghurt displayed higher radical inhibition than the controls. Considering the ABTS assay (Figure 7B), the addition of *M. oleifera*, either the extract or powder, improved the antioxidant capacity of the yoghurts against this radical throughout the eight weeks of the study. The milk protein hydrolysis that occurs during yoghurt production can contribute to the antioxidant properties presented [67]. Nevertheless, this study demonstrated that the incorporation of *M. oleifera* improved the antioxidant potential of the yoghurts, mainly due to their phenolic composition. Furthermore, all fortified yoghurts presented better antioxidant properties than PC yoghurt, except in the DPPH assay for t_2_ and t_3_ (four and eight weeks after production, respectively), where only ME2 presented better results. These results demonstrated the great potential of the use of *M. oleifera* powder and extract as natural antioxidants that can be incorporated into food to replace the synthetic compounds typically used. 

The antibacterial activity of the yoghurts was analysed against Gram-negative bacteria *E. coli* and Gram-positive bacteria *S. aureus*, and the results are expressed in Table 6. The incorporation of *M. oleifera* in the yoghurt did not increase its antibacterial activity, which is in accordance with the characterization of the extract, which did not present antibacterial activity against these bacteria. Also, a decrease was observed in the antibacterial activity against both *E. coli* and *S. aureus* over time. Furthermore, yoghurts demonstrated higher antibacterial capacity against *S. aureus* than *E. coli*. These results may be explained by the optimum pH conditions at which these bacteria grow: 6.0–7.0 for *E. coli* and 7.0–7.5 for *S. aureus* [68]. Since all yoghurt formulations presented a pH inferior to 4.1, these results were expected. 

Hereupon, these findings suggest that, although in the concentrations tested the addition of *M. oleifera* leaf extract and powder to the yoghurt did not change their antibacterial activity, the incorporation of these natural phenolic-reach additives improve the antioxidant properties of the product. 

#### 2.3.3. Microbial Analysis

The presence of microorganisms in food can impair its properties and threaten human health. Therefore, the microbial analysis of food products is extremely important as a check for the presence of microorganisms. The European Commission defined 10^2^ CFU/g as the limit of microorganisms that can be found in yoghurts to assure consumer safety [69]. In this work, the presence of coliform microorganisms was accessed using LSA medium while the presence of yeast and moulds was evaluated using RBC medium, four and eight weeks after the production of the yoghurts (t_2_ and t_3_, respectively). No microorganisms were detected, in both media, even after eight weeks of storage (0 CFU/mL). These findings may suggest that the formulations’ acidic pH generates an adverse environment for microbial growth. Furthermore, since the results were below the legal limits, the inclusion of *M. oleifera* had no impact on the safety of the consumers. 

## 3. Materials and Methods

### 3.1. Samples and Reagents

#### 3.1.1. Samples

Small branches with fresh mature leaves were collected from three *Moringa oleifera* trees in Luanda, Angola (8°57′24.9″ S, 13°13′02.9″ E). Upon arrival at the laboratory, the branches were washed with running tap water and the leaves were removed. The leaves were spread out on trays to dry, beginning at room temperature (about 30 °C) and ending in an oven at 50 °C until they were a constant weight. After this drying process, which lasted about two weeks, the leaves were crispy and ready for grinding. The grinding was done in a coffee grinder, using small amounts of leaves and the same grinding speed to obtain a fine powder with the same granulometry, with a particle size < 250 μm.

To produce yoghurt, UHT semi-skimmed milk and yoghurt with probiotics were purchased from a supermarket in Porto, Portugal. 

#### 3.1.2. Reagents

To analyse the total phenolic content, Folin–Ciocalteu reagent (Ref. 47641), from Sigma-Aldrich (St. Louis, MO, USA), and sodium carbonate (Ref. 1.06392, CNa_2_O_3_, CAS 497-19-8), from Merck (Darmstadt, Germany), were used.

To evaluate the antioxidant capacity, 2,2-diphenyl-1-picrylhydrazyl (Ref. D9132, C_18_H_12_N_5_O_6_, CAS 1898-66-4) and 2,2′-azino-bis(3-ethylbenzothiazoline-6-sulfonic acid) (Ref. A1888, C_18_H_24_N_6_O_6_S_4_, CAS 30931-67-0) were purchased from Sigma-Aldrich (St. Louis, MO, USA).

For the antibacterial analysis, agar (Ref. J637, CAS 9002-18-0) and Plate Count Agar (Ref. 84608.0500) were obtained from VWR (Radnor, PA, USA), Rose-Bengal Chloramphenicol Agar (Ref. 1.00467.0500) was acquired from Merck (Darmstadt, Germany) and m-Lauryl Sulfate Broth (Ref. 0734) was purchased from Sigma-Aldrich (St. Louis, MO, USA). 

Sorbic acid (Ref. S1626, C_6_H_8_O_2_, CAS 110-44-1), used as a control in yoghurt production, was acquired from Sigma-Aldrich (St. Louis, MO, USA).

For extraction and analysis, dimethyl sulfoxide (Ref. 41640, C_2_H_6_OS, CAS 67-68-5), from Honeywell (Charlotte, NC, USA), and ethanol (Ref. 83813.360, C_2_H_6_O, CAS 64-17-5), acetic acid (Ref. 20104.312, C_2_H_4_O_2_, CAS 64-19-7) and acetonitrile (Ref. 83639.320, C_2_H_3_N, CAS 75-05-8) obtained from VWR (Radnor, PA, USA) were used. 

Milli-Q water was purified with a water purification equipment, with 18.2 Ω of electrical resistance (Millipore, Burlington, MA, USA).

#### 3.1.3. Analytical Standards for HPLC-DAD Analysis

For analytical purposes, 7 flavonoids and phenolic acids were used: caffeic acid (Ref. C0625, C_9_H_8_O_4_, CAS 331-39-5), (+)-catechin hydrate (Ref. C1251, C_15_H_14_O_6_, CAS 225937-10-0), chlorogenic acid (Ref. C3878, C_16_H_18_O_9_, CAS 327-97-9), (−)-epicatechin (Ref. E1753, C_15_H_14_O_6_, CAS 490-46-0), gallic acid (Ref. 91215, C_7_H_6_O_5_, CAS 149-91-7), kaempferol (Ref. 60010, C_15_H_10_O_6_, CAS 520-18-3) and quercetin (Ref. Q4951, C_15_H_10_O_7_, CAS 117-39-5) were purchased from Sigma-Aldrich (St. Louis, MO, USA). All standards used were HPLC grade.

### 3.2. Extraction of Bioactive Compounds from Moring Leaf Powder

An ultrasound-assisted solid-liquid extraction method was used to extract the phenolic compounds from *M. oleifera* leaf powder. The samples were mixed with 70% ethanol in a ratio of 1 g of sample to 40 mL of solvent and placed in a J. P. Selecta 3000617 ultrasonic bath (50/60 kHz, 420 W, Barcelona, Spain) for 30 min at room temperature. Afterwards, the mixture was placed in a water bath under agitation (250 rpm) at 50 °C for 2.5 h to complete the extraction. Finally, the samples were filtered using a Whatman No. 1 paper and the solvent was completely evaporated using a rotary evaporator Rotavapor R-200 (BUCHI Laboratories, Switzerland), followed by lyophilisation. The extracts were stored at −4 °C. The extractions were conducted in triplicate to determine the extraction yield using Equation (1):Extraction Yield (%) = (m_extract_/m_sample_) × 100(1)
where m_extract_ is the mass of the extract obtained and m_sample_ is the mass of *M. oleifera* leaf powder used in the extraction.

### 3.3. Characterization of M. oleifera Extract

#### 3.3.1. Total Phenolic Content

To quantify the total phenolic content (TPC) of the extracts, the Folin–Ciocalteu method was used. First, 20 μL of the extract (1 g/L in ethanol) was mixed with 1580 μL of water and 100 μL of Folin–Ciocalteu reagent. After 3–6 min, 300 μL of sodium carbonate (333.3 g/L in water) was added and the mixture was incubated for 2 h in the dark at room temperature. Finally, the absorbance was analysed at 750 nm using a Thermo GENESYS™ 10UV UV-Vis spectrophotometer (Thermo Fisher Scientific, Waltham, MA, EUA) [70]. All measurements were performed in triplicate. The results were expressed in mg gallic acid equivalents (GAE)/g dried extract, after preparing a gallic acid calibration curve (0.5–10 mg/L).

#### 3.3.2. Antioxidant Capacity

The antioxidant capacity of the extract was analysed using the 2,2-diphenyl-1-picrylhydrazyl (DPPH) assay, according to the literature, with some modifications [71]. Briefly, 20 μL of the extract (0.1–2.5 g/L in ethanol) was incubated with 180 μL of a DPPH solution (150 μmol/L in ethanol) for 40 min in the dark, at room temperature. Then, the absorbance was analysed at 515 nm. All measurements were performed in triplicate. The inhibition percentage of DPPH was calculated using Equation (2), where Abs_sample_ is the absorbance of the extract, Abs_blank_ is the absorbance of 20 μL of water with 180 μL of ethanol and Abs_control_ is the absorbance of 20 μL of water with 180 μL of DPPH solution, after incubation. Finally, the IC_50_ value of the extract was determined using a calibration curve of the percentage of DPPH inhibition *versus* the extract concentration.
DPPH inhibition (%) = (1 − (Abs_sample_ − Abs_blank_)/(Abs_control_ − Abs_blank_)) × 100(2)

The 2,2′-azino-bis(3-ethylbenzothiazoline-6-sulfonic acid) (ABTS) assay was also used to analyse the extract’s antioxidant capacity, following the literature protocol [72]. First, 20 μL of the extract (0.1–1.0 g/L in ethanol) was incubated with 180 μL of ABTS reactive solution for 15 min at room temperature, protected from the light. Afterwards, the absorbance was analysed at 734 nm. All measurements were performed in triplicate. The percentage of inhibition of ABTS radical was determined using Equation (3), where Abs_sample_ is the absorbance of the extract and Abs_control_ is the absorbance of 20 μL of 0.05M acetate buffer (pH 4.6) with 180 μL of ABTS reactive solution, after incubation. Finally, the IC_50_ value of the extract was determined using a calibration curve of the percentage of ABTS inhibition vs. the extract concentration.
ABTS inhibition (%) = (Abs_control_ − Abs_sample_)/Abs_control_ × 100(3)

#### 3.3.3. Antibacterial Activity

The antibacterial activity of the extract was analysed by the disk diffusion test. Extract solutions (0.5 g/mL and 1.0 g/mL) were prepared in 2% aqueous dimethyl sulfoxide (DMSO). Bacterial suspensions of *Escherichia coli* and *Staphylococcus aureus* were prepared, with an optical density of 0.1 at 610 nm, and plated in Plate Count Agar (PCA) medium. Afterwards, sterile disks were added to the plates and 7 µL of the extract solution was added to the disks in triplicate. Ultrapure water was used as the negative control and sorbic acid as the positive control. The plates were incubated for 24 h at 37 °C and the diameter of the inhibition halos was measured [73].

#### 3.3.4. Analysis of Phenolic Compounds by HPLC-DAD

To identify and quantify the phenolic compounds of the extract, high-performance liquid chromatography (HPLC) was performed using an Elite LaChrom HPLC system (Hitachi, Japan), equipped with a Hitachi L-2200 autosampler, L-2130 pump, and L-2455 diode array detector (DAD). A Puroshper^®^ STAR RP-18 endcapped LiChroCART^®^ chromatography column (Merck, Germany) was used. Acetonitrile:water:ethanol (2:1:1 *v*/*v*/*v*) was used as a solvent to prepare standards and samples solutions. The sample solution was prepared by resuspending the extract obtained in 10 mL of the solvent. As eluents, Milli -Q water with 0.5% of orthophosphoric acid was used for the mobile phase A and methanol:acetonitrile (80:20 *v*/*v*) was used for the mobile phase B. The gradient was as follows: 0–10 min, 10% B; 10–25 min, 15% B; 25–40 min, 30% B; 40–50 min 50% B; 50–60 min, 70% B. The eluent flow rate and the injection volume were 0.8 mL/min and 40 μL, respectively. Analytes were identified by their retention time (RT) and measured accordingly to their maximum absorption wavelength: catechin and epicatechin—222 nm; gallic acid—275 nm; caffeic acid and chlorogenic acid—322 nm; kaempferol and quercetin—365 nm. Calibration curves were prepared and phenolic compounds were quantified by the external standard method. 

### 3.4. Incorporation of M. oleifera Extract in Yoghurt

#### 3.4.1. Yoghurt Production

The yoghurt was produced as described in the literature, with slight modifications [66]. Briefly, 1 L of UHT semi-skimmed milk was heated to 40 °C. Then, 125 mg of a commercial yoghurt with probiotics was added and the mixture was homogenised using a glass rod. Finally, the mixture was incubated at 37 °C for 16 h and then stored at 4 °C. Different formulations of yoghurt were produced and the additives used are described in Table 7. Sorbic acid was used as the positive control since it is a commonly used additive in yoghurts. *M. oleifera* extract and leaf powder were used as natural preservatives. The extract was added at the same level and twice the concentration of sorbic acid (formulations ME and ME2, respectively). The leaf powder added to the yoghurt (formulation MP) was equivalent to the extract added to formulation ME. The physicochemical characteristics, stability, and microbiological safety of the yoghurts were analysed for 8 weeks with four timepoints: t_0_—after production; t_1_—two weeks after production; t_2_—four weeks after production; and t_3_—eight weeks after production.

#### 3.4.2. pH Determination

To determine the pH value of the yoghurts produced, they were dissolved in ultrapure water (1:9 m/V). The mixture was homogenised using a T18 Digital Ultra-Turrax (IKA, Staufen, Germany) for 1 min at 300 rpm and the pH of the samples was determined using a digital pH meter [66]. All measurements were performed in triplicate.

#### 3.4.3. Syneresis

The protocol to determine the syneresis of the samples, which represents the amount of whey expelled from the yoghurt, was adapted from the literature [74]. Briefly, 3 g of yoghurt was centrifuged for 20 min at 700 rpm. Then, the supernatant was discarded and the precipitate was weighed. Equation (4) was used to determine the syneresis of the samples:Syneresis (%) = (m_supernantant_/m_yoghurt_) × 100(4)
where m_supernatant_ is the difference between the initial mass of the yoghurt and the mass of the precipitate and m_yoghurt_ is the initial mass of the yoghurt.

#### 3.4.4. Water Holding Capacity

The water holding capacity (WHC) was determined according to the literature [75]. For that, 7.5 mL of distilled water was added to 0.25 g of each yoghurt. The samples were vortexed for 1 min and then centrifuged for 20 min at 3000 rpm. Afterwards, the supernatant was discarded and the precipitate was collected and weighed. The precipitate was dried for 5 h at 105 °C and the WHC was determined using Equation (5):WHC = (m_fresh precipitate_ − m_dried precipitate_)/m_dried precipitate_(5)
where m_fresh precipitate_ is the mass of the precipitate before drying and m_dried precipitate_ is the mass of the precipitate after drying.

#### 3.4.5. Viscosity

To study the viscosity of the yoghurts, the apparent viscosity (mPa·s) was measured using an MCR 92 rheometer (Anton Paar, Graz, Austria). The apparent viscosity was analysed for different shear rates (0.01–100 s^−1^) at a constant temperature (25 °C). The effect of the temperature on the apparent viscosity of the samples was also analysed by varying the temperature from 2 °C to 25 °C and then back to 2 °C again, at a constant shear rate (50 s^−1^).

#### 3.4.6. Total Phenolic Content and Antioxidant Capacity

To analyse the antioxidant properties of the yoghurts, an extraction of the phenolics was performed. For that, 8 mL of ethanol were added to 2 g of each sample. The mixture was vortexed for 1 min, followed by an ultrasonic bath for 5 min. This process was repeated twice more. Finally, the samples were centrifuged at 3000 rpm for 20 min and the supernatant was collected and stored at 4 °C, in the dark, for further analysis of the TPC and antioxidant capacity (DPPH and ABTS) of the yoghurts, according to the protocols described in Section 3.3.1 and Section 3.3.2, respectively.

#### 3.4.7. Antibacterial Activity

To analyse the antimicrobial activity of the yoghurts, the well diffusion assay was used, which is similar to the disk diffusion assay described in Section 3.3.3. but with slight modifications. Instead of sterile disks, small wells were made in the PCA plates with a glass Pasteur pipette. Each sample was added to a well in triplicate. The diameter of the inhibition halos was measured after incubating the plates for 24 h at 37 °C. *E. coli* and *S. aureus* were again the model bacteria chosen to analyse the antimicrobial activity of the yoghurts and the bacterial suspensions were prepared as explained above.

#### 3.4.8. Microbial Safety

For this analysis, Lauryl Sulphate Agar (LSA) and Rose Bengal Chloramphenicol Agar (RBC) were used since they are selective to coliform microorganisms, and to yeast and moulds, respectively. The yoghurts were serially diluted in a saline solution (1:10 *v*/*v*) two times. Then, 100 μL of each solution was plated in both mediums. The LSA plates were incubated at 37 °C for 24 h, while the RBC plates were incubated for 7 days at 25 °C. Afterwards, the plates were checked for the presence of microorganisms.

### 3.5. Statistical Analysis

For the statistical analysis, an analysis of variance (ANOVA) was performed using the Tukey’s multiple comparisons test. GraphPad Prism 8.0.2 was used and values with *p* < 0.05 (95% confidence interval) were considered statistically different.

## 4. Conclusions

The purpose of this study was to evaluate the use of natural ingredients obtained from *Moringa oleifera* leaves in food fortification. A good extraction yield was obtained using ultrasound-assisted solid-liquid extraction, and the obtained extract presented antioxidant properties, with higher activity against ABTS compared to DPPH. However, no antibacterial activity was detected against *Escherichia coli* and *Staphylococcus aureus*. Seven main phenolic compounds were identified in the extract, namely caffeic, chlorogenic and gallic acids, catechin, epicatechin, kaempferol and quercetin. These results demonstrated that *M. oleifera* leaf extract has an interesting phenolic composition and, consequently, antioxidant capacity. This is an important property for its potential incorporation in food products that are susceptible to oxidation, such as the case of yoghurt. The yoghurts fortified with *M. oleifera* powder or extract presented slightly better stability than the negative control and improved antioxidant properties compared to both negative and positive controls. An increase in the extract concentration increased the total phenolic content of the formulation but reduced its stability regarding syneresis and water holding capacity. Comparing the leaf powder with the extract, the first presented a lower reduction in the TPC after eight weeks but also poorer stability regarding syneresis. All formulations demonstrated higher antibacterial activity against *S. aureus* than *E. coli* and no microbial contamination was detected after eight weeks of storage. The obtained results demonstrate that both *M. oleifera* powder and extract can be incorporated into yoghurt, creating a fortified food product that can be used to combat children’s malnutrition in developing countries. Even though it is outside of the scope of this work, toxicity studies are recommended on yoghurts produced with *Moringa oleifera* powder or extracts to ensure the safety of the product. Finally, a sensory analysis should also be carried out to understand the effects of the addition of these compounds on the sensorial properties of the yoghurts.

## Figures and Tables

**Figure 1 molecules-28-02526-f001:**
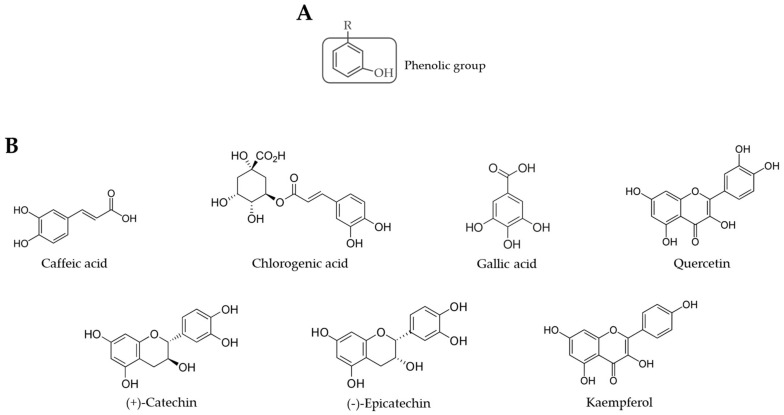
Chemical structure of a phenolic compound (**A**) and of the phenolics commonly identified in *M. oleifera* leaves (**B**).

**Figure 2 molecules-28-02526-f002:**
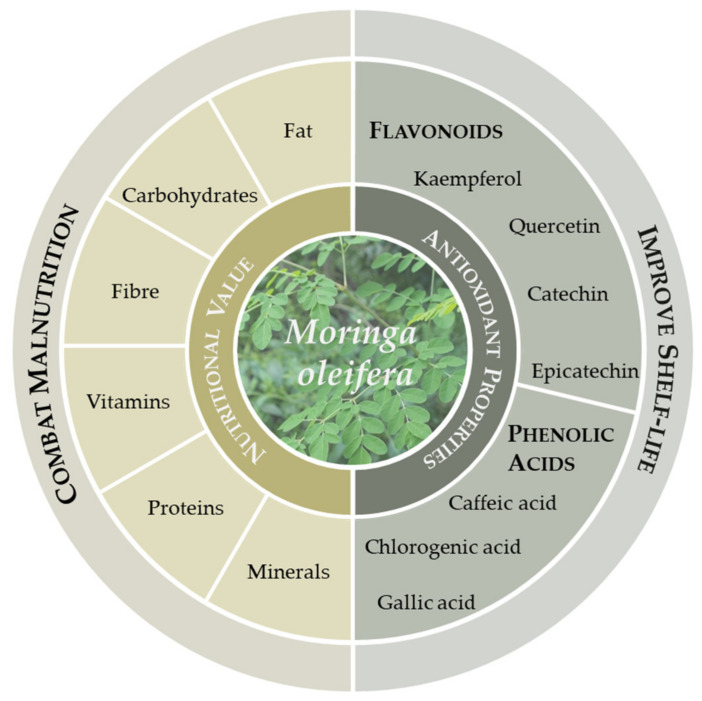
*Moringa oleifera* properties useful to food fortification.

**Figure 3 molecules-28-02526-f003:**
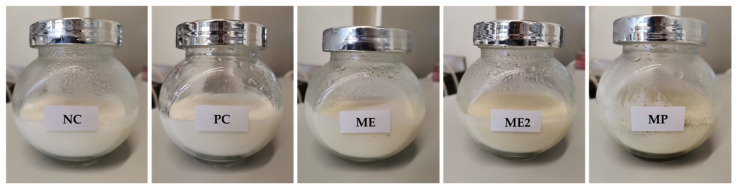
Yoghurt formulations appearance after production. NC—yoghurt with no additives (negative control); PC—yoghurt with sorbic acid (positive control); ME—yoghurt with 1 g/L of *M. oleifera* extract; ME2—yoghurt with 2 g/L of *M. oleifera* extract; MP—yoghurt with 2.9 g/L of *M. oleifera* leaf powder.

**Figure 4 molecules-28-02526-f004:**
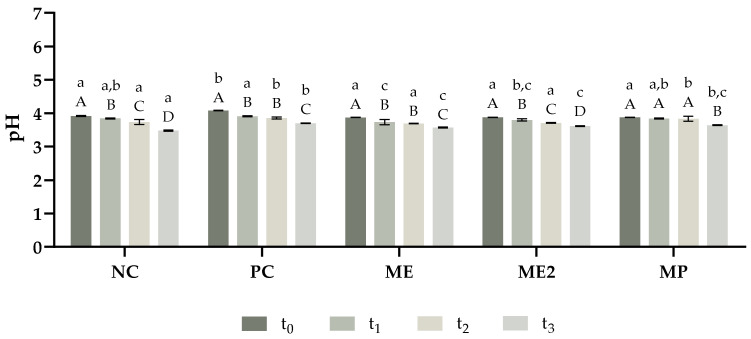
pH variation of the yoghurts throughout the period of the study. The analysis was performed in different timepoints: t_0_—after production; t_1_—two weeks after production; t_2_—four weeks after production; and t_3_—eight weeks after production. NC—yoghurt with no additives (negative control); PC—yoghurt with sorbic acid (positive control); ME—yoghurt with 1 g/L of *M. oleifera* extract; ME2—yoghurt with 2 g/L of *M. oleifera* extract; MP—yoghurt with 2.9 g/L of *M. oleifera* leaf powder. The results are expressed as means ± standard deviations of 3 independent measurements obtained from the same sample. Different lowercase letters (a–c) represent statistically different values (*p* < 0.05) for the same timepoint. Different capital letters (A–D) represent statistically different values (*p* < 0.05) for the same yoghurt.

**Figure 5 molecules-28-02526-f005:**
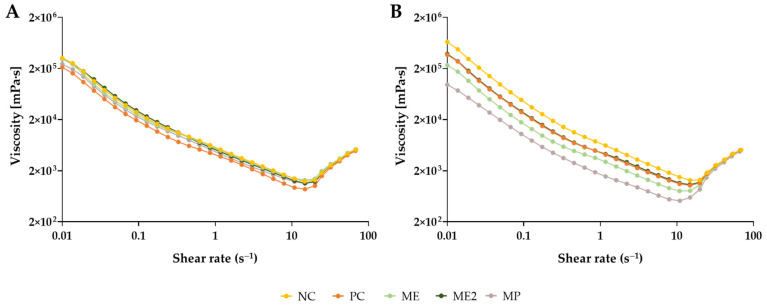
Viscosity variation of the yoghurts in function of the shear rate. The analysis was performed after production, t_0_ (**A**) and eight weeks after production, t_3_ (**B**). NC—yoghurt with no additives (negative control); PC—yoghurt with sorbic acid (positive control); ME—yoghurt with 1 g/L of *M. oleifera* extract; ME2—yoghurt with 2 g/L of *M. oleifera* extract; MP—yoghurt with 2.9 g/L of *M. oleifera* leaf powder.

**Figure 6 molecules-28-02526-f006:**
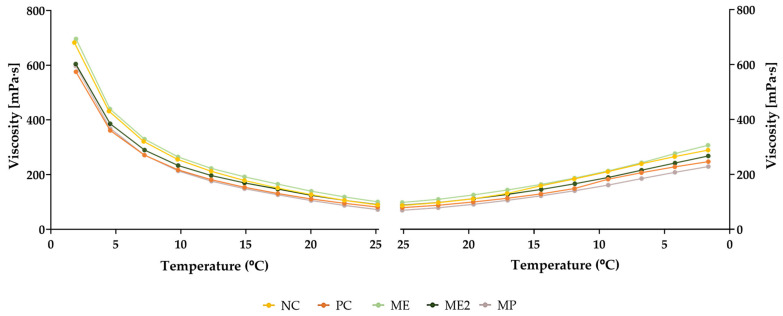
Viscosity variation of the yoghurts as a function of the temperature, after production (t_0_). NC—yoghurt with no additives (negative control); PC—yoghurt with sorbic acid (positive control); ME—yoghurt with 1 g/L of *M. oleifera* extract; ME2—yoghurt with 2 g/L of *M. oleifera* extract; MP—yoghurt with 2.9 g/L of *M. oleifera* leaf powder.

**Figure 7 molecules-28-02526-f007:**
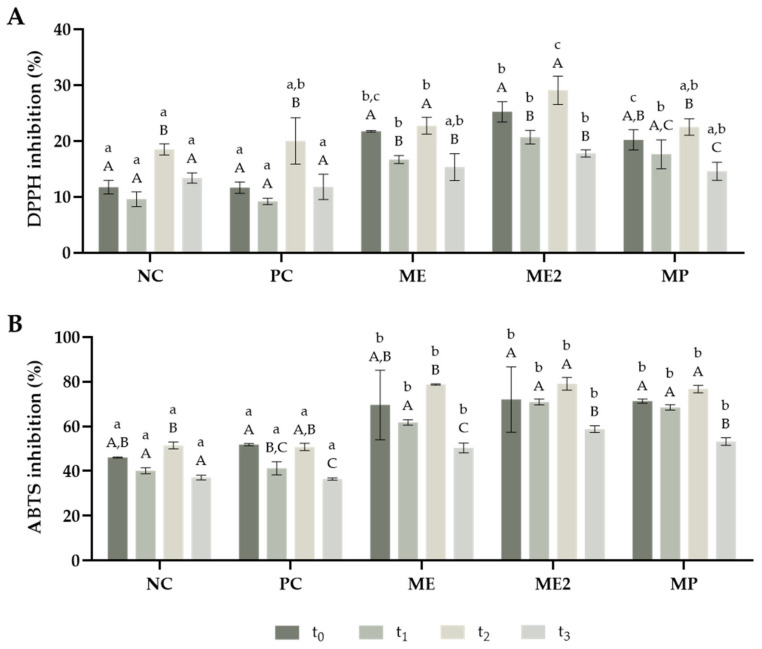
Antioxidant capacity of the yoghurts obtained for DPPH (**A**) and ABTS (**B**) throughout the period of the study. The analysis was performed in different timepoints: t_0_—after production; t_1_—two weeks after production; t_2_—four weeks after production; and t_3_—eight weeks after production. NC—yoghurt with no additives (negative control); PC—yoghurt with sorbic acid (positive control); ME—yoghurt with 1 g/L of *M. oleifera* extract; ME2—yoghurt with 2 g/L of *M. oleifera* extract; MP—yoghurt with 2.9 g/L of *M. oleifera* leaf powder. The results are expressed as means ± standard deviations of 3 independent measurements obtained from the same sample. Different lowercase letters (a–c) represent statistically different values (*p* < 0.05) for the same timepoint. Different capital letters (A–C) represent statistically different values (*p* < 0.05) for the same yoghurt.

**Table 1 molecules-28-02526-t001:** Studies on the incorporation of *Moringa oleifera* leaves on dairy products.

Food Product	Objectives	Results	Ref.
Powdered milk	Evaluate the efficacy of MO leaves as powdered milk to use as a supplementary food for malnutrition.	An increase in weight was observed in the children regularly supplemented with MO powdered milk for two months, compared to the control group.	[43]
Butter	Study the use of MO leaf extract for the stabilization of butter at refrigeration temperature.	The addition of the extract (600 ppm) did not impact the butter composition and improved the antioxidant properties of the product without impairing the overall acceptability.	[44]
	Improve the nutritional value of buttermilk using MO dry leaves to prevent and correct malnutrition.	The fortified buttermilk did not present any significant differences in pH or acidity but presented an increase in protein and ash content and also in vitamin C and vitamin B. The addition of 2% MO leaves did not alter the overall acceptability.	[45]
Cheese	Improve the nutritional value of cape gooseberry Petit Suisse cheese using MO leaf powder.	The addition of 2% MO to the formulation increased its nutritional value (increased ash, protein, fat, and fibre contents) but decreased the sensory acceptance.	[46]
	Improve the quality and shelf life of cream cheese using ethanolic MO leaf extract.	The addition of MO (up to 4%) increased the protein, ash and total phenolic contents, and also the antioxidant activity of the cheese. It also enhanced the growth of probiotic strains. An improvement was observed in both flavour and taste during storage.	[47]
Ice cream	Study the use of ice cream enriched with MO leaf powder as an alternative to sugar-sweetened ice cream.	The MO-enriched ice creams (0.05% and 0.5%) presented an increase in the total phenolic and flavonoid contents, and in the inhibition of α-amylase and α-glucosidase enzymes, and improved antioxidant properties. These ice creams reduced the glycaemic index in vitro but presented a reduction in the overall acceptability.	[48]
	Evaluate the influence of MO leaves-enriched ice cream on the redox and chlorogenic systems of rats.	The addition of MO to ice cream (0.59–2.35%) reduced the rat’s body weight gain, the glycaemic index and the lipid profile (triglycerides and cholesterol), inhibited the brain cholinergic enzymes (AChE and BChE) and increased the brain antioxidant enzyme activities (SOD and CAT). The overall acceptability of MO-enriched ice cream was lower than the control (commercial ice cream).	[49]
Yoghurt	Evaluate the effects of MO leaf extract on the fermentation, bioactive properties and quality characteristics of yoghurt.	The addition of the extract accelerated yoghurt fermentation by promoting the growth of lactic acid bacteria and increased the viscosity and free radical-scavenging during storage. Changes in the colour of the yoghurt were observed but the overall acceptability was not significantly influenced by the addition of 0.5% MO extract.	[50]
	Produce a yoghurt supplemented with MO leaf powder.	The fortified yoghurt presented a decrease in the syneresis and moisture, and an increase in total solids, protein and ash contents. The best results were obtained by combining 1% MO leaf powder and mango flavour.	[51]
	Develop a fortified yogurt with MO leaves as a carrier of probiotics and micronutrients.	The supplementation with MO (around 30%) increased the viability of the probiotic strain (*L. rhamnosus* GR-1) but reduced the overall acceptability of the product.	[52]

MO—*Moringa oleifera*; AChE—acetylcholinesterase; BChE—butyrylcholinesterase; SOD—superoxide dismutase; CAT—catalase.

**Table 2 molecules-28-02526-t002:** Results from the bioactive characterization of *M. oleifera* extract.

TPC (mg_GAE_/g_dried extract_)	Antioxidant Capacity (IC_50_—mg_extract_/L)		Antibacterial Activity (d_halo_—mm)	
	DPPH	ABTS	*E. coli*	*S. aureus*
54.5 ± 16.8	133.4 ± 12.3	60.0 ± 9.9	ND	ND

ND—not detected; TPC—total phenolic content; GAE—gallic acid equivalents; IC_50_—concentration of extract needed to inhibit 50% of the free-radicals; DPPH: 2,2-diphenyl-1-picrylhydrazyl; ABTS—2,2′-azino-bis(3-ethylbenzothiazoline-6-sulfonic acid). The results are expressed as means ± standard deviations of three independent measurements obtained from the same sample.

**Table 3 molecules-28-02526-t003:** Main phenolic compounds of *M. oleifera* extract quantified by HPLC-DAD.

Compound	RT (min)	Calibration Curves	R^2^	IDL (mg/L)	IQL (mg/L)	Standard Concentration (mg_compound_/g_extract_)
Caffeic acid	29.23	A = 5.56 × 10^5^ C − 1.56 × 10^6^	0.9992	4.03	13.43	0.16
Catechin	24.38	A = 1.57 × 10^5^ C − 7.93 × 10^5^	0.9861	41.50	138.34	19.83
Chlorogenic acid	26.62	A = 1.91 × 10^5^ C − 1.16 × 10^5^	0.9999	2.84	9.48	1.04
Epicatechin	30.34	A = 4.15 × 10^5^ C − 1.56 × 10^6^	0.9983	5.82	19.39	0.67
Gallic acid	11.28	A = 1.21 × 10^5^ C + 1.33 × 10^6^	0.9978	27.96	93.20	0.13
Kaempferol	52.79	A = 7.34 × 10^5^ C + 2.41 × 10^5^	0.9993	1.49	4.97	0.02
Quercetin	49.34	A = 7.37 × 10^5^ C − 2.68 × 10^5^	0.9994	1.37	4.58	0.06

RT—retention time; A—peak area; C—standard concentration (mg/L); R^2^—coefficient of determination; IDL—instrumental detection limit; IQL—instrumental quantification limit.

**Table 4 molecules-28-02526-t004:** Consistency index (K) and flow consistency index (n) parameters obtained for yoghurts.

Yoghurt	t_0_		t_3_	
	K (mPa·s^n^)	n	K (mPa·s^n^)	n
NC	9322	0.451	12135	0.366
PC	6729	0.486	8275	0.431
ME	8984	0.469	6036	0.478
ME2	9096	0.438	8545	0.431
MP	8020	0.476	3745	0.548

NC—yoghurt with no additives (negative control); PC—yoghurt with sorbic acid (positive control); ME—yoghurt with 1 g/L of *M. oleifera* extract; ME2—yoghurt with 2 g/L of *M. oleifera* extract; MP—yoghurt with 2.9 g/L of *M. oleifera* leaf powder.

**Table 5 molecules-28-02526-t005:** Total phenolic content variation of the yoghurts throughout the period of the study.

Yoghurt	TPC (mg/L)			
	t_0_	t_1_	t_2_	t_3_
NC	0.54 ± 0.01 ^a,A^	0.54 ± 0.01 ^a,A^	0.38 ± 0.01 ^a,B^	0.20 ± 0.02 ^a,C^
PC	0.66 ± 0.02 ^a,A^	0.61 ± 0.01 ^a,A^	0.44 ± 0.02 ^a,B^	0.32 ± 0.03 ^a,C^
ME	0.88 ± 0.08 ^b,A^	0.65 ± 0.03 ^a,B^	0.60 ± 0.06 ^b,B^	0.45 ± 0.12 ^b,C^
ME2	0.93 ± 0.05 ^b,A^	0.79 ± 0.03 ^b,B^	0.71 ± 0.04 ^b,c,B^	0.57 ± 0.04 ^b,c,C^
MP	0.88 ± 0.06 ^b,A^	0.85 ± 0.03 ^b,A^	0.85 ± 0.03 ^c,A^	0.63 ± 0.01 ^c,B^

NC—yoghurt with no additives (negative control); PC—yoghurt with sorbic acid (positive control); ME—yoghurt with 1 g/L of *M. oleifera* extract; ME2—yoghurt with 2 g/L of *M. oleifera* extract; MP—yoghurt with 2.9 g/L of *M. oleifera* leaf powder. The results are expressed as means ± standard deviations of 3 independent measurements obtained from the same sample. Different lowercase letters (a–c) represent statistically different values (*p* < 0.05) in the same column. Different capital letters (A–C) represent statistically different values (*p* < 0.05) in the same row.

**Table 6 molecules-28-02526-t006:** Antibacterial activity of the yoghurts, for t_0_ and t_2_, against *E. coli* and *S. aureus*.

Yoghurt	*E. coli*		*S. aureus*	
	t_0_	t_2_	t_0_	t_2_
NC	10.7 ± 0.5	ND	14.3 ± 0.9 ^A^	9.0 ± 1.6 ^B^
PC	11.0 ± 1.4 ^A^	8.3 ± 0.9 ^B^	17.7 ± 3.7 ^A^	10.0 ± 0.8 ^B^
ME	11.7 ± 0.9	ND	15.3 ± 1.7 ^A^	9.0 ± 0.8 ^B^
ME2	12.3 ± 0.9 ^A^	8.7 ± 0.5 ^B^	16.3 ± 0.9 ^A^	11.0 ± 0.8 ^B^
MP	11.7 ± 0.9 ^A^	7.5 ± 0.5 ^B^	15.3 ± 0.5 ^A^	10.7 ± 1.2 ^B^

NC—yoghurt with no additives (negative control); PC—yoghurt with sorbic acid (positive control); ME—yoghurt with 1 g/L of *M. oleifera* extract; ME2—yoghurt with 2 g/L of *M. oleifera* extract; MP—yoghurt with 2.9 g/L of *M. oleifera* leaf powder. The results are expressed as means ± standard deviations of 3 independent measurements obtained from the same sample. Different letters (A, B) represent statistically different values (*p* < 0.05) in the same row.

**Table 7 molecules-28-02526-t007:** Additives incorporated in the different yoghurt formulations produced.

Formulation	Additives
NC	yoghurt with no additives (negative control)
PC	yoghurt with 1.0 g/L * of sorbic acid (positive control)
ME	yoghurt with 1.0 g/L of *M. oleifera* extract
ME2	yoghurt with 2.0 g/L of *M. oleifera* extract
MP	yoghurt with 2.9 g/L * of *M. oleifera* leaf powder

* Legal limit of sorbic acid in yoghurts.

## Data Availability

Not applicable.

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
