# Peer review of "Incorporation of Moringa oleifera Leaf Extract in Yoghurts to Mitigate Children’s Malnutrition in Developing Countries"

_molecules, 2023, doi:10.3390/molecules28062526_

Round 1

Reviewer 1 Report

The manuscript titled "Incorporation of Moringa oleifera Leaf Extract in Yoghurts to Mitigate Children’s Malnutrition in Developing Countries”. Although there are articles on this topic, the authors give a description of a specific application. However, the manuscript should be improved.

Introduction:

Line 32: It is important that the authors mention the production (kg or ton per year) of this plant.

Line 40: The authors should mention which is the concentration of total polyphenols and antioxidant capacity present in leaves as well as which are the specific families of polyphenols present in the leaves. For example, flavanols, stilbenes…

Line 42 – 44: the authors show nutritional benefits present in the leaves, but they don´t show the problems related with the presence of toxic compounds and their effects on human health.

Line 46 - 50: Although authors mention that Moringa Oleifera is rich in protein, polysaccharides, proteins, and vitamins, they don’t mention its content. In addition, they should show other sources like vegetables and fruits in order to compare the chemical composition and bioactive compounds.

Line 53 – 65: Although authors mention several works on the incorporation of Moringa oleifera leaves in dairy products, they do not highlight what is pending to be resolved or studied. This would allow them to improve their objective.

Materials and methods

Line 331-332: If polyphenols are oxidized in convective drying processes, why did the authors use 50°C?

Line 334: What was the particle size obtained?

Line 360: Authors should mention in the text the purity of the standards used

Line 420: The authors quantified only 7 polyphenols. So, they should justify in the introduction why these polyphenols are the most important. On the other hand, what were the equations of the calibration curves, the R2 values and the detection limits?

Line 445: How do the authors establish the levels of powder and leaf extract?

If the addition of polyphenols can generate bitterness and astringency in yogurt; Why didn't the authors carry out a sensory evaluation test?

Line 502: The authors should detail the type of statistical design and comparison test.

Results

Line 79 – 82: The paragraph should go in the introduction.

Line 84 – 112: Although the authors used an extraction condition, TPC, IC50, and ABTS must be compared with other works to see similarities or differences as well they should explain why these differences.

Line 113 – 120: In the HPLC analysis, the authors should explain why catechin is the major component, as well as compare the results with other works.

Line 121 – 125: The paragraph should go in the introduction.

Table 3: Usually the specific polyphenol content is expressed in ug/g. However, the authors report values in mg/g. So what is the justification for this result?

Figure 1: It does not correspond to the results section

Line 129 – 132: It should be in the conclusions section.

Line 154 – 155: Can other microorganisms also affect pH?

Line 174: Authors should explain in detail the type of interaction between casein and whey proteins.

Figure 5 should show the statistical analysis

Line 247 – 312: The authors should improve the presentation of results. For example, treatment x was 2 times better than treatment, as well as they should explain in detail why the differences.

The abstract and conclusions should be improved according to the previous recommendations.

In conclusion, there are aspects that must be improved in this work.

Author Response

Comments and Suggestions for Authors

The manuscript titled "Incorporation of Moringa oleifera Leaf Extract in Yoghurts to Mitigate Children’s Malnutrition in Developing Countries”. Although there are articles on this topic, the authors give a description of a specific application. However, the manuscript should be improved.

Introduction:

Line 32: It is important that the authors mention the production (kg or ton per year) of this plant.

Answer: The authors did not find any information of the production of M. oleifera per year worldwide. Regarding the production per plant, it depends on the particular conditions (soil and climatic conditions, plant density) in each harvest site. For the location and harvest of this study, 333 g of dry leaf matter per plant was obtained. In another location, with different soil characteristics, 495 g of dry leaf matter per plant was obtained. Once the leaves are harvested, they immediately regrow and three months later they can be harvested again.

Line 40: The authors should mention which is the concentration of total polyphenols and antioxidant capacity present in leaves as well as which are the specific families of polyphenols present in the leaves. For example, flavanols, stilbenes…

Answer: Thank you for your comment. The information was added to the manuscript, as suggested (lines 43-53).

Line 42 – 44: the authors show nutritional benefits present in the leaves, but they don´t show the problems related with the presence of toxic compounds and their effects on human health.

Answer: Thank you for your comment. The information was added to the manuscript, as suggested (lines 65-76).

Line 46 – 50: Although authors mention that Moringa Oleifera is rich in protein, polysaccharides, proteins, and vitamins, they don’t mention its content. In addition, they should show other sources like vegetables and fruits in order to compare the chemical composition and bioactive compounds.

Answer: Thank you for your comment. The information was added to the manuscript, as suggested (lines 59-64).

Line 53 – 65: Although authors mention several works on the incorporation of Moringa oleifera leaves in dairy products, they do not highlight what is pending to be resolved or studied. This would allow them to improve their objective.

Answer: Thank you for your comment. The information was highlighted in the manuscript, as suggested (lines 101-103).

Materials and methods

Line 331-332: If polyphenols are oxidized in convective drying processes, why did the authors use 50°C?

Answer: The final drying step in an oven at 50 ºC, which lasted no more than 48 hours, was necessary for the leaves to become crispy. Other investigators reported that oven drying at 60 ºC of pretreated M. oleifera Leaves did not result in any significant change in the nutrients in comparison with wind and sun drying (Arun PR, Rajan AP, Santhalia S (2011). Comparative analysis of preservation techniques on Moringa oleifera. Inter.J. Agric. and Food Sci. 1: 12-22).

Line 334: What was the particle size obtained?

Answer: The information was added to the manuscript (line 428).

Line 360: Authors should mention in the text the purity of the standards used

Answer: Thank you for your comment. All standards used were HPLC grade. The specific purity of each standard can be accessed on the manufacturer's website by searching for the standard’s reference mentioned in the article (in brackets).

Line 420: The authors quantified only 7 polyphenols. So, they should justify in the introduction why these polyphenols are the most important. On the other hand, what were the equations of the calibration curves, the R2 values and the detection limits?

Answer: Thank you for your comment. The authors quantified the polyphenols most commonly found in Moringa oleifera leaves, as stated in the introduction (lines 43-45). The information regarding the equations of the calibration curves, the R2 values and the detection limits were added to Table 3 in the Results section of the manuscript.

Line 445: How do the authors establish the levels of powder and leaf extract?

Answer: The leaf extract level was established by the legal limit of the positive control. The legal limit of sorbic acid (positive control) in yoghurts is 1.0 g/L. Therefore, leaf extract was added at the same level (formulation ME) and with twice the legal limit of sorbic acid (formulation ME2). The quantity of powder to be used was determined taking into consideration the extraction yield (around 34%, i.e. 100 g of powder originate 34 g of extract). Therefore, the quantity of powder used was equivalent to the quantity of extract used in formulation ME. This information was added to Section 3.4.1. of the manuscript.

If the addition of polyphenols can generate bitterness and astringency in yogurt; Why didn't the authors carry out a sensory evaluation test?

Answer: Thank you for your comment. The authors understand the importance of performing a sensory analysis of the product, to analyse the impact of the addition of Moringa oleifera-derived additives on the sensory properties of yoghurts. However, the primary purpose of this work was to understand the potential properties of the additives studied that can be advantageous for their incorporation into yoghurts. The authors decided that it was better to perform a sensory analysis in future work when the yoghurt’s formulation is optimized. The authors included a statement in the Conclusions sections regarding the importance of performing such analysis in future work. Nevertheless, some laboratory members have volunteered to taste the produced yoghurts and find them a pleasant product.

Line 502: The authors should detail the type of statistical design and comparison test.

Answer: Thank you for your comment. Some information was added to Section 3.5. of the manuscript, as suggested.

Results

Line 79 – 82: The paragraph should go in the introduction.

Answer: Thank you for your comment. The alteration was made, as suggested.

Line 84 – 112: Although the authors used an extraction condition, TPC, IC50, and ABTS must be compared with other works to see similarities or differences as well they should explain why these differences.

Answer: Thank you for your comment. The suggested information was added to Section 2.2. of the manuscript.

Line 113 – 120: In the HPLC analysis, the authors should explain why catechin is the major component, as well as compare the results with other works.

Answer: Thank you for your comment. The suggested information was added to Section 2.2. of the manuscript.

Line 121 – 125: The paragraph should go in the introduction.

Answer: Thank you for your comment. The alteration was made, as suggested.

Table 3: Usually the specific polyphenol content is expressed in ug/g. However, the authors report values in mg/g. So what is the justification for this result?

Answer: Although it may be more common to express the results in μg/g, other studies had already expressed the phenolic content in mg/g. Also, the authors believe that the units in mg/g are more suitable for the results obtained in this work.

Figure 1: It does not correspond to the results section.

Answer: Thank you for your comment. Figure 1 was placed in the Introduction section.

Line 129 – 132: It should be in the conclusions section.

Answer: Thank you for your comment. The alteration was made, as suggested.

Line 154 – 155: Can other microorganisms also affect pH?

Answer: The lactic acid bacteria are responsible for the lactic acid fermentation that reduces the pH of the yoghurt. Other microorganisms can also change the pH of their environment, like sulfate-reducing bacteria. However, those microorganisms are generally not found in yoghurt. Also, since no contaminations were detected after eight weeks of storage, the authors believe that lactic acid bacteria were the only microorganisms present in yoghurt.

Line 174: Authors should explain in detail the type of interaction between casein and whey proteins.

Answer: Thank you for your comment. More information about the interaction between casein and whey proteins was added to Section 2.3.1. of the manuscript, as suggested.

Figure 5 should show the statistical analysis

Answer: Thank you for your comment. The authors did not present the statistical analysis for these assays because independent measures were not performed. However, these results provide a tendency for the yoghurts’ behaviour through time regarding syneresis and WHC.

Line 247 – 312: The authors should improve the presentation of results. For example, treatment x was 2 times better than treatment, as well as they should explain in detail why the differences.

Answer: Thank you for your comment. Some alterations were made in the Results section of the manuscript, as suggested.

The abstract and conclusions should be improved according to the previous recommendations.

Answer: Thank you for your comment. Some alterations were done in the Abstract and Conclusions sections, as suggested.

In conclusion, there are aspects that must be improved in this work.

Reviewer 2 Report

This manuscript reported the incorporation of M. oleifera leaf extract in yoghurts by the evaluation of the effect of M. oleifera leaf powder and extract on the stability, antioxidant, and antimicrobial activities of the yoghurt. This topic is interesting. However, more scientific discussions and comparison with literatures will be possibly beneficial to highlight the merits of this study. Furthermore, some issues that need to be addressed are listed in the following.

(1) The graphical abstract cannot highlight the findings of this study, and it is unnecessary to be presented before the introduction section.

(2)  Section 2.1. What are the functions of ultrasound and subsequent agitation in the extraction of bioactive compounds? Please discuss in more detail for the roles of ultrasound and agitation as well as the duration on the extraction yield. The comparison of extraction yield between the literatures and this study is also suggested.

(3) Section 2.2, lines 92-95. The descriptions are not clear. How do the extraction method and conditions influence TPC? What is the meaning of “the values range”?

(4)  Figures 4 and 8. Do the standard deviations of n = 3 obtained from the results of three independent measurements of a sample or from results of three independent samples? This point for statistical analysis should be clarified.

(5) Figure 5. It is suggested to perform the statistical analysis for the results of WHC. A more detailed discussion between ME and ME2 is required

(6)  The quality of Figures 6 and 7 should be improved.

(7)  Table 5. What are the reasons for the different reducing rates of TPC over time between ME and MP? Do the probiotics in the yoghurt affect the TPC of ME or MP during eight weeks of storage?

(8)  Section 3.1.1, line 336. What is the full name of “UHT”? What are the probiotics and their amounts in the yoghurt?

(9) Sections 3.1.2 and 3.1.3. Please check and correct the address of the manufacturers of the chemical reagents.

(10)  Section 3.2. What are the frequency and power of the ultrasonic bath? What is the agitation speed? The information about the model of the facilities and city, state, and country of the manufacture should be provided.

(11)  Section 3.4.5. Please check the unit of shear rate.

(12)  Section 3.4.6, line 486. Where are the sections 4.3.1 and 4.3.2?  

(13)  The normal decimal points should be used in digits of results in all Tables.

Author Response

Author's Reply to the Review Report (Reviewer 2)

Comments and Suggestions for Authors

This manuscript reported the incorporation of M. oleifera leaf extract in yoghurts by the evaluation of the effect of M. oleifera leaf powder and extract on the stability, antioxidant, and antimicrobial activities of the yoghurt. This topic is interesting. However, more scientific discussions and comparison with literatures will be possibly beneficial to highlight the merits of this study. Furthermore, some issues that need to be addressed are listed in the following.

(1) The graphical abstract cannot highlight the findings of this study, and it is unnecessary to be presented before the introduction section.

Answer: Thank you for your comment. Although the graphical abstract is not a mandatory section of the manuscript, the authors believe that it is an important topic to be included in the article since it helps the readers to easily comprehend the purpose and goals of the developed work as well as the carried out work to achieve that purpose. The authors believe that the graphical abstract presented does not highlight any findings of the study but it does give information about the objective of the work (create a fortified yoghurt to combat children’s malnutrition) and what was done to achieve it (extraction of compounds of interest and characterization of the extract and the fortified yoghurts). Finally, since the journal template and author’s guidelines do not provide a specific topic/place to exhibit the graphical abstract, the authors believe that this should be presented after the abstract since it resumes the work done in the paper.

(2)  Section 2.1. What are the functions of ultrasound and subsequent agitation in the extraction of bioactive compounds? Please discuss in more detail for the roles of ultrasound and agitation as well as the duration on the extraction yield. The comparison of extraction yield between the literature and this study is also suggested.

Answer: Thank you for your comment. The suggested information was added to Section 2.1.

(3) Section 2.2, lines 92-95. The descriptions are not clear. How do the extraction method and conditions influence TPC? What is the meaning of “the values range”?

Answer: Thank you for your comment. The information was clarified, as suggested (lines 119-138).

(4)  Figures 4 and 8. Do the standard deviations of n = 3 obtained from the results of three independent measurements of a sample or from results of three independent samples? This point for statistical analysis should be clarified.

Answer: The standard deviations were obtained from the results of three independent measurements of a sample. The information was added to the legends of Figures 4 and 8.

(5) Figure 5. It is suggested to perform the statistical analysis for the results of WHC. A more detailed discussion between ME and ME2 is required

Answer: Thank you for your comment. The authors did not present the statistical analysis for these assays because independent measures were not performed. However, these results provide a tendency for the yoghurts’ behaviour through time regarding syneresis and WHC. A more detailed discussion between ME and ME2 was added, as suggested (lines 268-271).

(6)  The quality of Figures 6 and 7 should be improved.

Answer: Thank you for your comment. The quality of Figures 6 and 7 was improved.

(7)  Table 5. What are the reasons for the different reducing rates of TPC over time between ME and MP? Do the probiotics in the yoghurt affect the TPC of ME or MP during eight weeks of storage?

Answer: When ingested, the majority of the phenolic compounds can not be absorbed in the small intestine, reaching the colon. There, they need to be metabolized by gut microbiota microorganisms. Lactobacillus bulgaricus and Streptococcus thermophilus are the main probiotics used in the production of yoghurt and can be found in the human gut microbiota. Therefore, these probiotics may be able to initiate biotransformation of the phenolic compounds, which may explain the reduction of the TPC over time. A brief sentence regarding this topic was added to Section 2.3.2. of the manuscript. Regarding the different reducing rates of TPC between ME and MP, the authors believe that phenolic compounds can be less exposed in the leaves powder than in the extract, which may confer them more protection from degradation over time.

(8)  Section 3.1.1, line 336. What is the full name of “UHT”? What are the probiotics and their amounts in the yoghurt?

Answer: UHT means ultra-high temperature. The yoghurt’s manufacturer does not specify which probiotics are present in the yoghurt, nor their amount.

(9) Sections 3.1.2 and 3.1.3. Please check and correct the address of the manufacturers of the chemical reagents.

Answer: Thank you for your comment. The addresses of the manufacturers were revised and corrected.

(10)  Section 3.2. What are the frequency and power of the ultrasonic bath? What is the agitation speed? The information about the model of the facilities and city, state, and country of the manufacture should be provided.

Answer: Thank you for your comment. The suggested information was added to Section 3.2. of the manuscript.

(11)  Section 3.4.5. Please check the unit of shear rate.

Answer: Thank you for your comment. The units of shear rate were corrected.

(12)  Section 3.4.6, line 486. Where are the sections 4.3.1 and 4.3.2? 

Answer: Section 3.4.6. Total phenolic content and antioxidant capacity (line 579) is followed by sections 3.4.7. Antibacterial activity (line 587), 3.4.8. Microbial safety (line 595), and finally section 3.5. Statistical analysis (line 602), in the methods. These are followed by section 4. Conclusions (line 607).

(13)  The normal decimal points should be used in digits of results in all Tables.

Answer: Thank you for your comment. All Tables were revised and the suggested alterations were made.

Round 2

Reviewer 1 Report

In my opinion, the manuscript has been improved, whose changes and their results may be useful to other researchers working in this field. I recommend the publication of the manuscript.

Author Response

The authors wish to express their appreciation to the Reviewer for their valuable comments on the manuscript.

Reviewer 2 Report

There are three issues (points 4, 5, and 10) that the authors did not solve in the revised version, listing in the following:

(4)  In Figures 4 and 8, it is not proper for data comparison using standard deviations (n=3) estimated from the results of three independent measurements of a single sample. It is essential to estimate the standard deviation from the data of independent samples for the statistical analysis in the bio-systems.

(5)  In Figure 5, the statistical analysis of WHC is required for the clear comparison in significant difference.

(10) Section 3.2. Are you sure that the ultrasonic frequency of the ultrasonic bath is 50/60 Hz?

Author Response

Manuscript ID: molecules-2216973

Author's Reply to the Review Report (Reviewer 2)

Comments and Suggestions for Authors

There are three issues (points 4, 5, and 10) that the authors did not solve in the revised version, listing in the following:

(4)  In Figures 4 and 8, it is not proper for data comparison using standard deviations (n=3) estimated from the results of three independent measurements of a single sample. It is essential to estimate the standard deviation from the data of independent samples for the statistical analysis in the bio-systems.

Answer: Thank you for your comment. The authors understand the importance of performing independent measures of independent samples. However, such analysis was not performed in this study and, therefore, it is not possible to calculate the standard deviation from the data of independent samples. The authors will take this into consideration in further studies.

(5)  In Figure 5, the statistical analysis of WHC is required for the clear comparison in significant difference.

Answer: Thank you for your comment. The authors understand the importance of performing statistical analysis of the results. However, triplicate measures were not performed, therefore, it is impossible to perform statistical analysis. The authors will take this into consideration in further studies.

(10) Section 3.2. Are you sure that the ultrasonic frequency of the ultrasonic bath is 50/60 Hz?

Answer: The ultrasonic frequency of the ultrasonic bath is 50/60 kHz. The information was corrected in the manuscript (line 466).
